

# Deep-sea search and recovery: with and without operating an underwater vehicle

Tongwei Zhang[1,2,3], Shengjie Qin[1], Xiangxin Wang[1], Jialing Tang[1]

[1]National Deep Sea Center, Qingdao 266237, China
[2]Laboratory for Marine Geology, Qingdao National Laboratory for Marine Science and Technology, Qingdao, 266061, China
[3]Joint Laboratory for Ocean Observation and Detection, Qingdao National Laboratory for Marine Science and Technology, Qingdao, 266237, China

*Correspondence to*: Tongwei Zhang (tongwei.zhang@outlook.com)

**Abstract.** Deep-sea search and recovery mainly refers to the search, recovery, and salvage of objects with high value that are lost on the deep-sea bottom. Deep-sea search and recovery objects include aircraft black boxes, underwater vehicles, and other types of objects. The recovery and salvage of objects involves accurately obtaining their underwater positions. Depending on whether or not the salvage object carries an acoustic beacon, two methods are available: onboard acoustic signal search and near-bottom sweep search and search. Once the underwater position of a salvage object is known, it can be recovered and salvaged with a remotely operated underwater vehicle (ROV) and/or human-occupied vehicle (HOV). However, there are many difficulties with the practical application of existing deep-sea recovery systems that are based on the deep-sea operation of ROVs and HOVs. Based on the design idea and working mode of TV-grab in oceanography, this paper proposes a new type of deep-sea recovery system that does not rely on operating underwater vehicles and presents its recovery process. The new deep-sea recovery system combines underwater optical imaging, mechanical docking/grasping, acoustic imaging and positioning, and propeller operating to provide low-cost and rapid deep-sea recovery. Compared to the deep-sea recovery system with a ROV and/or HOV, the new deep-sea recovery system without an operating underwater vehicle described in this paper is proposed to be used, but not tested yet.

## 1 Introduction

Traditionally, recovery and salvage at sea refers to handling emergencies in offshore and shallow-water areas (water depth of several hundred meters), especially in navigation channels. However, deep-sea areas with a depth of more than 2000 m account for 84% of the oceans. Therefore, most of Earth's surface is covered by deep sea. As human activities continue to expand into deep seas, such as the airline crashes of Air France 447 and Malaysia Airlines MH370, new challenges are arising for deep-sea search and recovery (Bian et al., 2016; Breivik and Allen, 2008; Coppini et al., 2016; Qin, 2014; Wang, 2013; Zhang and Li, 2017).





Deep-sea search and recovery refers to the search, recovery, and salvage of objects with high value (e.g., aircraft black boxes, underwater vehicles, and crashed vehicles) that are lost on the seabed. The search and recovery of underwater objects has two main steps. In the search step, the underwater position of the salvage object needs to be accurately obtained as a prerequisite for underwater recovery. Depending on whether the salvage object carries acoustic beacons, the underwater search

can be divided into two types: onboard acoustic signal search and near-bottom sweep search and search. Once the underwater position of the salvage object is accurately known, it can be recovered and salvaged with a deep-sea recovery system in the recovery step.

Since the 1960s, deep-sea remotely operated underwater vehicles (ROVs) and human-occupied vehicles (HOVs) have played an important role in accident recovery and salvage with significant social influence. For example, Air France Flight

447 crashed in the Atlantic Ocean in 2009. In 2011, the Woods Hole Oceanographic Institution (WHOI) used the side-scan sonar carried by the REMUS 6000 autonomous underwater vehicle (AUV) to discover most of the wreckage of the aircraft at the bottom of the Atlantic Ocean at a depth of 3900 m. The recovery of the aircraft's black box was completed by Phoenix International's Remora 6000 ROV.

However, because of the high cost of use and maintenance of deep-sea ROVs and HOVs and their limited operating

depth, it is difficult for them to recover and salvage underwater objects that lack social influence but still have high value, such as the Kaiko ROV lost in 2003 (Hashimoto et al., 2004), Autosub-2 AUV lost in 2005 (Strutt, 2006), ABE AUV lost in 2010, and Nereus HROV lost in 2014 (Showstack, 2014). Because lost underwater vehicles cannot be retrieved from the deep sea or under ice, the cause of their failure can only be speculated upon according to existing evidence and cannot be verified. Only the salvage of these wrecked underwater vehicles can truly clarify the cause of the accident. This has important scientific

significance and engineering practical value for improving their safety and reliability. Therefore, there is an urgent need for a low-cost deep-sea recovery system.

This paper introduces the basic types of deep-sea search and recovery and analyzes the advantages and difficulties of each type. It then introduces the basic methods and equipment for deep-sea search. Next, the salvage of submarine objects using ROV/HOV is discussed. A new type of deep-sea recovery system is proposed that does not rely on any vehicles operating

underwater. The paper ends with some conclusions.

## 2 Deep-sea search and recovery objects

Deep-sea salvage objects mainly include aircraft black boxes, damaged underwater vehicles (e.g., ROVs and AUVs), and other objects lost on the seabed (e.g., deep-sea landers, aircraft, and military targets). Most salvage objects do not weigh very much in water (<1000 kg).



## 2.1 Aircraft black boxes

An obvious advantage of conducting submarine searches for aircraft black boxes is that such objects are equipped with underwater positioning beacons, whose acoustic signals can be picked up by the shipboard acoustic positioning system. When an aircraft crashes in a vast ocean, the seabed search for the black box faces several great difficulties. First, obtaining a relatively accurate crash location is difficult, which makes the search scope very large. For example, researchers analyzed the "handshake signal" between aircraft and satellites and between satellites and ground receiving stations to determine the area where Malaysia Airlines MH370 may have crashed. The area was approximately 60,000 km2 with an average water depth of 4000 m and maximum depth of more than 7800 m. Second, the lack of ocean-related data increases the search difficulty. For example, the Malaysian Airlines MH370 crash area did not have an accurate multi-beam topographic map. Instead, satellite altimetry was used to retrieve the topographic map of the seabed with limited accuracy.

The black box is a rectangular parallelepiped or cylinder with a length of approximately 50 cm, weight in water of approximately 10 kg, and orange-red surface. For a recovery system, the distinctive features of the aircraft black box are its small size and weight, which can be grasped by a manipulator. Therefore, when an aircraft black box position is determined, an ROV is usually dispatched to salvage it.

## 2.2 Underwater vehicles

Continuous developments in science and engineering technology have increased the operating depth and working time of various types of underwater vehicles (ROV/AUV). However, the unusual complexity of the water surface and underwater environment means that underwater vehicles inevitably experience various kinds of component failures (e.g., sensors, cables, software systems, actuators, and buoyancy materials). In general, these underwater vehicles operate at depths of several kilometers or under thick ice, so the failure of any part has catastrophic consequences.

Searching for underwater vehicles lost on the seabed has several advantages. (1) The locations of underwater accidents are known with relatively good accuracy because the motherships were in contact with the underwater vehicles before they were lost. (2) An accurate multi-beam submarine topography map is available because an underwater multi-beam sounding system is used to obtain underwater topographic maps of the area before the underwater vehicle dives. (3) Accurate hydrological data are available because the hydrological conditions of the dive zone are obtained by a shipborne temperature and saline depth survey (CTD) before the dive, and the ocean current information are obtained by the shipborne acoustic Doppler current profiler (ADCP).

Submarine search and recovery has several main disadvantages. (1) When underwater vehicles are lost at depths of several kilometers and even more than 10 km, there is usually a lack of corresponding recovery equipment. (2) When there is a lack or failure of a positioning beacon, the exact position of the salvage object is unknown.

If the buoyancy material of the underwater vehicle has imploded and causes the entire buoyancy material system to fail, the gravity of the entire system of the underwater vehicle is much greater than the buoyancy. For example, after Nereus HROV



lost its buoyancy material, it weighed about 1 T in the water, and when ABE lost its buoyancy material, it weighed about 200 kg in water. However, an operational ROV/HOV can provide a buoyancy load of around 200 kg, so using a deep-sea ROV/HOV to directly salvage underwater vehicles is not possible.

## 2.3 Others

The deep-sea lander can be equipped with a variety of sensors and cameras for long-term, fixed-point, and continuous in situ observations of the deep-seafloor and has been widely used in modern oceanic expeditions. In recent years, several deep-sea landers have been lost on the seabed. Because they record a large amount of valuable data, salvaging them would have great significance. These landers have a large number of hookable and force-receiving positions that can be remotely linked to recovery and salvage equipment through the design of suitable mechanical mechanisms.

Various crashed or retrievable aircraft (e.g., rockets, space shuttles, and intercontinental missiles) also fall in the deep ocean. If the weight of the wreck to be salvaged is very high, a neutral-buoyancy composite fiber rope may be needed. The breaking force of the rope is determined according to the weight of the salvage object, and the corresponding rope diameter is selected. A deep-sea ROV/HOV is used to hook the rope at the bottom. If the weight of the debris is not very large, a mechanical device like a television grab can be specifically designed to salvage it.

## 3 Underwater searches for salvage objects

### 3.1 Shipborne sound signal search

A shipboard sonar system can be used to search for the signals of underwater objects carrying acoustic beams. Once an aircraft black box enters the water, the water-sensitive switch on the carrying beacon activates the beacon to transmit a sound wave frequency of 37.5 kHz to the surrounding seawater through a metal shell. The sound source level is about 160 dB, the theoretical

maximum action distance is 4000–6000 m, and the beacon can last for 30 days. Specialized search and recovery vessels are usually equipped with sonar systems capable of locating 37.5 kHz acoustic beacons. If the surface search vessel does not detect the signal within 30 days, then the battery charge is gradually depleted, and the acoustic signal weakens until it stops working altogether. At this point, the corresponding acoustic positioning method is also no longer applicable.

      The underwater vehicles used in oceanic surveys generally use ultra-short baseline (USBL) for positioning. The operating

frequency of the acoustic localization beacon installed on the underwater vehicle is 10–16 kHz, and the working distance is generally about 8000 m. The beacon can be powered by a self-carried battery or the underwater vehicle. When a damaged underwater vehicle is being searched for, a query sound wave signal is transmitted through the USBL of a surface vessel, and an underwater response signal is transmitted after the underwater beacon is received. This can be used to accurately obtain the position of the underwater vehicle. If the beacon battery is dead, the beacon fails due to implosion, or the capabilities of USBL

are exceeded, then this acoustic positioning method will also fail.





### 3.2 Deep-sea terrain mapping

Accurate information on the seafloor topography is the supporting basis for the search and recovery of underwater objects. Although gravity maps can be used to rapidly invert a topographic map of the seabed, they have low accuracy and cannot meet the needs of deep-sea search and recovery. To map the seafloor topography at depths of several or even tens of kilometers, a large-area walking survey can be performed with a shipborne deep-sea multi-beam sounding system to obtain relatively accurate data.

Because of the vastness of oceans, shipborne deep-sea multi-beam measurement requires much time and money. Therefore, this method is mainly used along oceanic routes, in areas of high research interest (e.g., abysses, trenches, and cold springs), and in areas rich with oil, gas, and mineral resources (e.g., flammable ice, polymetallic nodules, cobalt-rich crusts, and hydrothermal vents). Therefore, after the scope of an underwater search and recovery is determined, the onboard deep-sea multi-beam sounding system should be used to map the area and obtain a relatively accurate topographic map of the seabed to provide basic terrain data.

### 3.3 Near-sea-bottom scan search

Although a shipborne deep-sea multibeam sounding system can detect a wide range of deep seabed terrains, its detection accuracy is limited, and it cannot search for small objects on the seafloor. A large-scale search of deep-sea salvage objects mainly relies on underwater vehicles such as the deep tow system and AUVs to carry sonar with side-scanning functions (e.g., side-scan sonar and sounding side-scan sonar).

### 3.3.1 Underwater mobile vehicles

The deep tow system can be equipped with various acoustic detection devices, such as side-scan sonar and shallow profiles. It is linked with the mothership through the armored photoelectric composite cable and can be towed at a height of tens of meters from the seabed (Cao et al., 2016). The scanning image of the side-scan sonar can be uploaded to the mothership in real time through the photoelectric hybrid cable to facilitate observation and judgment by the search personnel. Because the deep tow system is powered by a cable, it can operate continuously and for a long time underwater and is particularly suitable for a wide search range. The drawback is that it requires a large turning radius and has low mobility.

The AUV can achieve an equal height or depth cruise in underwater and completely autonomous detection according to a predetermined lateral line (Li, et al., 2016). Its advantages are a reduced reliance on the mothership and improved work efficiency. In contrast to the deep tow system, the scanned image obtained by the AUV cannot be uploaded to the mothership in real time. The detection results need to wait until the AUV is recovered to the deck and the data are post-processed. In addition, because the AUV is battery-powered, its underwater operating time is limited.





### 3.3.2 Side-scan sonar

Side-scan sonar transmits a sound wave signal and receives the reflected echo signal to form an acoustic image of the state of the sea bottom, including the position, current status, and height of the target object (Dong et al., 2009). Compared with other subsea detection technologies, side-scan sonar has the advantages of image visualization, high resolution, and large coverage.

Side-scan sonar on an underwater vehicle can reach a distance of tens of meters from the sea bottom. At low speeds, high-quality side-scan sonar images can be obtained, and even pipelines a dozen centimeters wide can be distinguished. Recently, some deep-tow side-scan sonar systems have been operated at high speeds, and high-resolution underwater images were obtained at 10 kn. This shows that side-scan sonar is particularly suitable for searching seabed targets.

Bathymetric side-scan sonar combines the side-sweep and multi-beam sounding technologies for synchronous

measurement of the seafloor topography. A new generation of bathymetric side-scan sonar has adopted high-resolution 3D acoustic imaging technology and the subspace fitting method to simultaneously measure multiple underwater targets and obtain high-resolution images from complex seabed and multipath signals (Sun et al., 2005).

### 4 Deep-sea recovery based on operational underwater robots

Operational ROVs and HOVs are deep-sea heavy equipment that carry robots and can perform underwater operations. The

existing deep-sea recovery and salvage system mainly depends on operational ROVs and HOVs (Liu, 2006).

### 4.1 Recovery process

First, the operational ROV/HOV needs to carry an USBL beacon. After it arrives at the position of the salvage object, it places the USBL beacon next to the object, and video is shot at multiple angles. Even when the location of the object has been obtained in advance, the deep-sea navigation accuracy is limited, and the underwater visible range is only 5–8 m. Thus, the

operational ROV/HOV may take a long time to find the salvage object. If a 3D real-time imaging sonar is installed on the ROV/HOV, the search time can be greatly reduced. For example, Echoscope's 3D real-time imaging sonar utilizes phased-array technology to generate 16, 000 simultaneous beams to form 3D sonar images and visualize the entire scene in real time. Its maximum range is 120 m, which is a qualitative leap compared to underwater visibility.

If the salvage object is lightweight, such as an aircraft black box, it can be directly grabbed by the deep-sea ROV/HOV

and carried back to the surface. If the salvage object is too heavy, then ropes or steel cables are required. In this case, the deep-sea ROV/HOV mainly performs work such as traction and hooking.

Scientific vessels are usually equipped with geological winches, conductivity–temperature–depth (CTD) winches, and so on. The geological winch can bear a load of up to 1 T. The CTD winch can bear a lower weight of several hundred kilograms. If an object weighing several tons needs to be recovered, then composite buoyant fiber ropes may be needed. The rope diameter

is determined according to the weight of the salvage object and the breaking force of the rope.

The specific salvage process is as follows:



(1)  Prior to salvage, the ROV/HOV is placed on the seabed.

(2)  USBL beacons and weights are installed at the ends of ropes/steel cables, and USBL on the mothership is used to localize the beacons beside the salvaged object, at the ends of the rope/steel cables, and on the ROV/HOV.

(3)  The ROV/HOV is maneuvered to guide the end of the rope/cable towards the salvage object.

(4)  The ROV/HOV attaches the ropes/cables to the salvage object by hooks.

(5)  After the ROV/HOV moves away from the salvage object, a USBL remote-controlled beacon at the end of the rope/cable releases weights.

(6)  The winch slowly recovers the rope/steel cable while USBL is used to monitor the depth of the beacon at the rope/steel cable end in real time.

(7)  After the salvage objects are recovered to the water surface, they are hoisted to the deck by a crane on the ship.

(8)  The ROV/HOV is finally recovered.

## 4.2 Typical recovery case

In general, a successful deep-sea recovery requires the cooperation of various types of underwater equipment to exploit their respective advantages. For example, in 1966, the US Navy's CURV I ROV cooperated with the Alvin HOV to recover a

hydrogen bomb that was lost at a depth of 914 m near Spain. After the US Challenger space shuttle explosion in 1986, manned and unmanned submersibles cooperated to recover 50 T of waste film and debris over a period of 6 months. This provided a reliable basis for an analysis of the Challenger space shuttle accident. In 1989, the former Soviet Union's Komsomol's nuclear submarine sank to a depth of 1860 m, and manned and unmanned submersibles again cooperated to sample seabed sediment, measure the radioactive dose, and seal submarine rafts.

The use of deep-sea ROVs/HOVs to salvage objects maximizes their advantages in large-scale operations, and their manipulators can be used flexibly. However, deep-sea ROVs/HOVs are large-scale deep-sea equipment. They have complex systems, require a special technical maintenance and protection team, and have a high diving cost. This makes them mainly suitable for accidents with significant social impact, such as the salvage of aircraft black boxes and debris. They are rarely used to salvage other lost objects.

**5 New type of deep-sea recovery system**

Based on the design idea and working mode of TV-grab in oceanographic studies, this paper proposes a new type of deep-sea recovery system that does not rely on any operational underwater vehicles and presents the recovery process. The new deep-sea recovery system combines underwater optical imaging, mechanical docking/grasping, acoustic imaging and positioning, and propeller driving to provide low-cost and rapid deep-sea recovery.



## 5.1 TV-grab

TV-grab is a visual grab sampler that combines the continuous observation of seabed images with a grab sampler (Zhang et al., 2005; Geng et al., 2009; Cheng et al., 2011), as shown in Figure 1. It is mainly composed of a sampler, armored cable, and shipboard control board. The sampler frame is equipped with a submarine television camera, light source, and power supply device. The sampler is connected to the onboard steering board and television display device through the armored coaxial cable, and a deep-sea winch is used to place the sampler 5–10 m from the seabed during operation. At this height, the sampler is dragged at a low speed of 1–2 kn and provides continuous observation of the seabed to find the target. Once the target is found, the sampler is dropped to the seafloor, and its claw is closed around the object. The sampling claw is opened and closed by an electric hydraulic manipulator inside the TV-grab. The outstanding features of the TV-grab are that they can observe the seafloor and record video directly and can accurately sample the target while being remote-controlled from the deck.

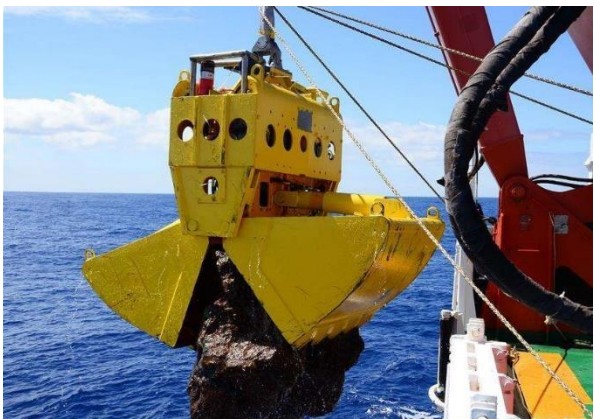

**Figure 1**. TV-grab.

## 5.2 New deep-sea recovery system

The new deep-sea recovery system combines a seabed optical camera, mechanical alignment and grabbing, acoustic imaging and positioning, and propeller drive for rapid deep-sea recovery on the seabed. Figure 2 shows a schematic of the new deep-sea recovery system.

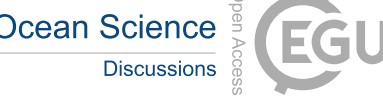



**Figure 2.** Schematic diagram of the new deep-sea recovery system.

Figure 3 shows the hardware block diagram of the new deep-sea recovery system. The onboard part mainly comprises a deck control system, a deck display system, a deck support system, a USBL, and external sensors. The underwater part includes

5    docking/gripping systems, propeller systems, optical camera systems, 3D acoustic imaging systems, underwater fiber optic





multiplexers, and USBL beacons. The armored photoelectric composite cable connects the onboard and underwater parts and serves as a power supply and communication medium and weight for the underwater portion. Most of the onboard part already exists on modern integrated ships. Only a few units such as the underwater docking/grasping control units, underwater propeller control units, and underwater optical/3D acoustic imaging display units need to be added.

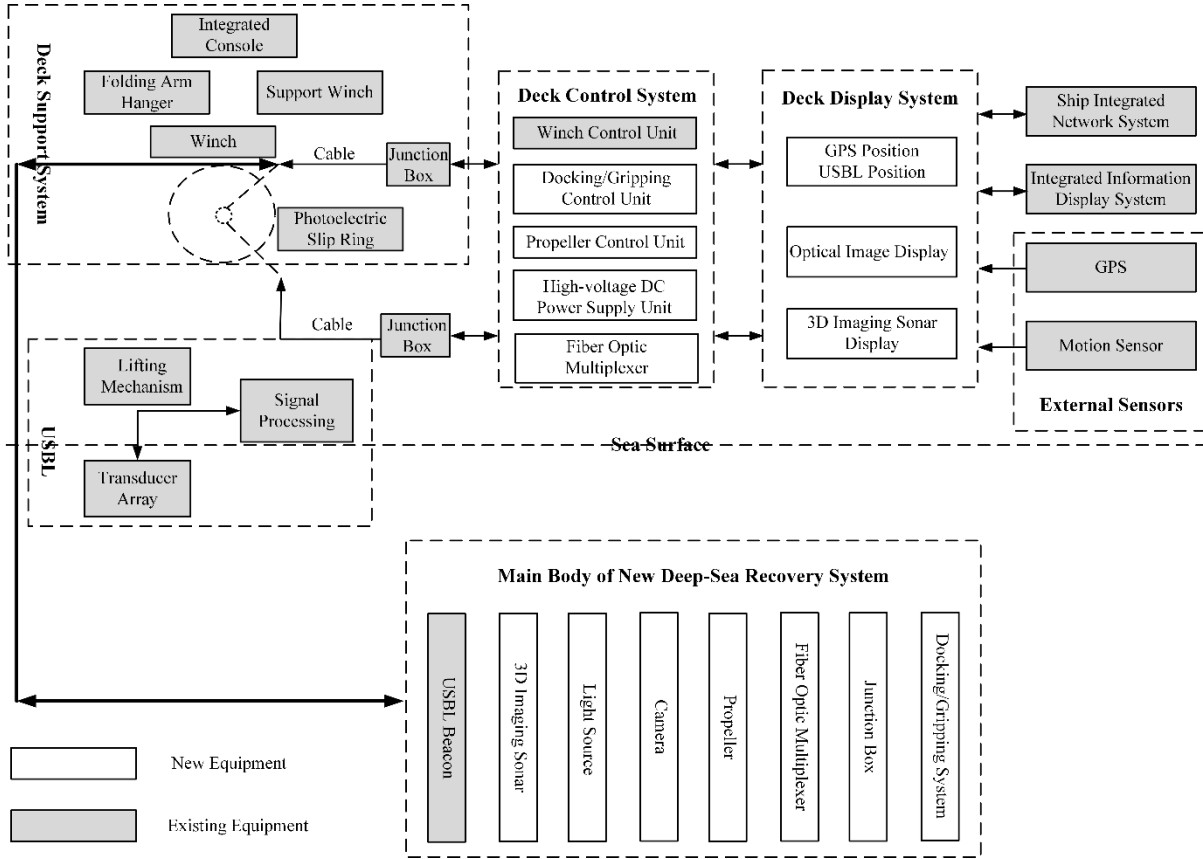

**Figure 3.** Hardware block diagram of the new deep-sea recovery system.

The deck control system is the overall control unit. It contains control units for the winch, underwater docking/gripping, underwater propeller, high-voltage DC power supply, and fiber optic multiplexer. It also provides the human–machine interface for controlling the underwater recovery system and power supply. The winch control unit is connected to the deck support system and controls the winch that extends and retracts cables. The underwater docking/grab control unit is connected to the underwater docking/gripping system by the armored photoelectric hybrid cable. The underwater propeller control unit is connected to the underwater propeller system by the armored photoelectric hybrid cable. The high-voltage DC power supply unit is connected to the underwater main body by the armored photoelectric composite cable. Fiber optic multiplexers are connected to the armored photoelectric composite cable.



The deck display system mainly displays the position, optical image, and 3D real-time sonar image for underwater search and docking. The position indicator is connected to the ship's Global Positioning System (GPS) and USBL via cable. The optical image display is connected to the underwater optical camera system through a fiber optic multiplexer. The 3D real-time imaging sonar display is connected to the underwater 3D real-time imaging sonar through a fiber optic multiplexer.

The deck support system includes the armored photoelectric composite cable, winches, photoelectric slip rings, guide pulleys, and folding arm hangers for the deployment and recovery of the underwater part. These components are connected to the deck control system and controlled by the winch control unit.

The USBL positioning system mainly includes a signal processing unit, lifting mechanism, and transducer array for real-time positioning and tracking of the underwater part of the recovery system. The signal processing unit is connected to an

external sensor to read the position, time, and attitude information. Through the cooperation of the response or trigger mode and the underwater USBL beacon, the deep-sea recovery system is positioned. The positioning results are displayed by the deck display system and are used to guide the approach to the salvage object.

External sensors mainly comprise the GPS and fiber optic motion sensor. GPS provides position and time information that is output via the cable to the USBL and position display. The fiber optic motion sensor provides attitude information and

is output via the cable to the USBL for position resolution.

The docking/gripping system is the core of the recovery system and includes an actuator and drive mechanism that support repeated opening and closing for connection with salvage objects. They are connected to the deck control system by the armored photoelectric composite cable and are controlled by the underwater docking/scraping control unit.

The propeller system includes multiple propellers for fine-tuning the relative positions of the recovery system and salvage

object on the horizontal plane to assist with docking and gripping (the relative positions are adjusted by retractable cables on the vertical plane). They are connected to the deck control system with the armored photoelectric composite cable and are controlled by the underwater propeller control unit.

The optical camera system consists of a light source and camera with a working distance of 5–8 m for observation during docking/snapping. It is connected to a deck display system through the armored photoelectric composite cable to display

images.

The 3D acoustic imaging system uses 3D real-time imaging sonar and is an important supplement to the optical camera system. It can perform 3D real-time imaging of underwater objects within a range of 120 m for early detection of salvage objects. It is connected to a deck display system by the armored photoelectric composite cable.

Underwater fiber optic multiplexers are paired with surface fiber optic multiplexers. One end is connected to the optical

camera and 3D acoustic imaging systems, and the other end is connected to the armored photoelectric composite cable for the transmission of optical and acoustic images.

The USBL beacon is matched with USBL for real-time positioning and tracking of the underwater part of the recovery system. If the transponder mode is used, the USBL beacon is relatively independent and powered by its own battery. If the trigger mode is used, a synchronous clock on the ship sends synchronization pulses to the signal processing units of the USBL





and USBL beacon through the coaxial cable and armored photoelectric composite cable, respectively. The USBL beacon can be powered externally. The advantages of the transponder mode are its simplicity and convenience, and the disadvantage is the large positioning interval. The advantages of the trigger mode are the single-path positioning and high update rate of the positioning data, and the disadvantage is the increased cost of the synchronization clock. Junction boxes are used to collect and transfer lines.

## 5.3 Recovery process

The new deep-sea recovery system can only be used when the position of the salvage object on the seabed is determined. Figure 4 shows the flowchart for deep-sea recovery with the new system.



**Figure 4.** Recovery flowchart with the new deep-sea recovery system.

The specific salvage process is as follows:



(1) The ship needs to navigate above the salvage object, and the dynamic positioning function is implemented if available. If no dynamic positioning is available, the ship centers on the position of the salvage object and floats at low speed. When further attention is needed, the top of the ship's bow should be selected in order to avoid accidents when the armored cable is pushed towards the ship bottom.

(2) The new recovery system is lowered into the water with the winch at a tapping speed of 40 m/min. When the height from the seafloor is less than 500 m, the cable release speed is reduced.

(3) The optical camera system and 3D real-time imaging sonar are turned on. The latter has a working distance of 120 m, so the salvage object can be discovered early. If the seabed terrain is complex, the 3D real-time imaging sonar should be set at a height of 30–50 meters from the recovery system. The recovery system and surrounding environment can be monitored without the seabed being touched.

(4) USBL is used to track the position of the new recovery system and guide it towards the salvage object.

(5) After the 3D real-time imaging sonar finds the salvage object, the guidance mode is switched from USBL to 3D real-time imaging sonar.

(6) After the optical camera system detects the salvage object, the underwater propeller control unit in the deck control system is used to fine-tune the horizontal position of the recovery system relative to the salvage object. The winch control unit slowly retracts and releases the cable to fine-tune the vertical position.

(7) When the docking/gripping system can dock/scrape the salvage object, the underwater docking/grasping control unit of the deck control system drives the actuator of the docking/gripping system to complete attachment.

(8) Docking/gripping is confirmed through the optical camera system. If the docking/gripping fails, then the underwater docking/scraping control unit of the deck control system is used to open the pick/grip actuator. Steps 6 and 7 are repeated until docking/crawl succeeds.

(9) The winch control unit of the deck control system slowly recovers the recovery system, and the optical camera system is used for real-time observation.

(10) When water is discharged, if the volume and weight of the salvage object are not large, it is directly recovered to the deck. If the salvage object is heavy or large in volume, it needs to be lifted to the deck by a folder or A-frame crane on the ship. In the middle, the ship releases a rubber boat to complete the crane hook.

In addition to better equipment for deep-sea recovery, recovery methods and operating techniques also need to be developed. The entire recovery system needs many people to work together, especially after the recovery system enters the water. Specific control personnel should master the system's performance and know how to use it properly. Recovery personnel should work closely with the ship's driving department to improve the recovery efficiency and avoid accidents.

Furthermore, compared to the deep-sea recovery system with a ROV and/or HOV, the new deep-sea recovery system without an operating underwater vehicle described in this paper is proposed to be used, but not tested yet.

## 6 Conclusions

A precondition for the recovery of underwater objects is to accurately obtain their position. The search and recovery area needs to be determined, and the salvage object needs to be found before the acoustic beacon is exhausted. If the sound beacon fails, underwater vehicles such as a deep-tow/AUV carrying side-scan sonar or bathymetric side-scan sonar need to be used to reach the seafloor and scan for the salvage objects.

Once the underwater position of the salvage object is accurately determined, an ROV/HOV is often used for recovery and salvage. However, there are many difficulties with the practical application of the existing deep-sea recovery system based on ROVs/HOVs. Based on the design idea and working mode of the TV-grab in oceanographic studies, this paper proposes a new type of deep-sea recovery system that does not rely on any vehicles operated underwater and presents the recovery process. The new deep-sea recovery system combines underwater optical imaging, mechanical docking/grasping, acoustic imaging and positioning, and propeller driving for low-cost and rapid deep-sea recovery. It can be used to salvage not only aircraft black boxes but also lost AUVs, ROVs, and landers.

**Competing interests.**

The authors declare that they have no conflict of interest.

**Acknowledgements.**

This work was supported in part by the National Natural Science Foundation of China under Grant 41641049, Qingdao National Laboratory for Marine Science and Technology under Grant QNLM2016ORP0406, Taishan Scholar Project Funding under Grant TSPD20161007, National Key R&D Program of China under Grant 2017YFC0305700, Shandong Provincial Natural Science Foundation under Grant ZR2015EM005,Shandong Provincial Key R&D Program under Grant 2016GSF115006, and Qingdao Independent Innovation Project under Grant 15-9-1-90-JCH.

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
