# Peer review of "Deep-sea search and recovery: with and without operating an underwater vehicle"

_Ocean Science, 2018_

## Short Comment (SC1) · 4 Nov 2018

The author of this manuscript provide a novel systematic framework for deep-sea search and recovery of the lost underwater target. The new recovery system includes TV-Grab, optical camera and acoustical imaging sonar, docking system. These combination can improve the flexibility and reliability of this recovery system, and this system is very useful and effective for small lost underwater target.

There is one point for attention. For your docking/grabing system, what is the maximum weight for lifting a lost target? This is a very important factor for whole systematic design, but this manuscript did not discuss it. Pleas add some comments.

By the way, language for the whole manuscript is needed to improve.

---

## Referee Comment (RC1) · Anonymous Referee #1 · 7 Nov 2018

As human activities continue to expand into deep seas, more requirements are needed for deep sea search and recovery. Currently, ROVs and HOVs are the main forces for deep sea recovery. Due to the high cost and scarcity of ROVs and HOVs, the authors in this paper proposed a new type of deep-sea recovery system based on TV-grab in oceanography. The quite detailed design and its recovery process are described. Although the new system has not been implemented and tested yet, based on this reviewer's experience, this concept is feasible and can easily be implemented. Therefore, the present concept is of some practical value and to those who have the TV-grabber but lack of ROV and HOV, they can use TV-grabber to carry out the recovery task in an emergency situation. The paper is in overall quite comprehensive and well presented. In terms of the originality, it is really not so much and in terms of

the economic benefit, it is also very limited since TV-grabber is a type of ROV and their operational costs are more or the less the same. In terms of the technical contents, this reviewer think it is not ocean science scope but ocean technology scope. Therefore, it is recommended for rejection in this journal but the authors are suggested to submit to the technology scope journal. Some other minor corrections and suggestions are given in the attached PDF file.

Please also note the supplement to this comment:
https://www.ocean-sci-discuss.net/os-2018-88/os-2018-88-RC1-supplement.pdf

[Figure]

**Supplement:**

[revised manuscript text omitted]

---

## Author Comment (AC1) · 13 Nov 2018

We thank the Reviewer for the comments, which have helped improve our manuscript.

The author of this manuscript provides a novel systematic framework for deep-sea search and recovery of the lost underwater target. The new recovery system includes TV-Grab, optical camera and acoustical imaging sonar, docking system. This combination can improve the flexibility and reliability of this recovery system, and this system is very useful and effective for the small lost underwater target.

Response: We thank the Reviewer for his overall positive assessment of our work.

There is one point for attention. For your docking/grabbing system, what is the maximum weight for lifting a lost target? This is a very important factor for whole systematic

design, but this manuscript did not discuss it. Please add some comments.

Response: We thank the Reviewer for this comment. The maximum weight for lifting a lost target mostly depends on the armored photoelectric composite cable. In the revised paper, we explicitly state that: "Most TV-grabs can sample up to 1000 kg or more at a time (Clark et al., 2016)." (Page 8, L9–10). Furthermore, as mentioned in Page 2, L29–30, "Most salvage objects do not weigh very much in water (<1000 kg)." Hence, in the revised paper, we explicitly state that: "The maximum weight of the new deep-sea recovery system for lifting a lost target is 1000 kg in water." (Page 8, L16–17). "Clark, M. R., Consalvey, M., Rowden, A. A.: Biological sampling in the deep sea, Wiley-Blackwell, New Jersey, 207-227pp., 2016." has been added to References.

By the way, language for the whole manuscript is needed to improve.

Response: We thank the Reviewer for this comment. The revised paper has been edited by Editage [www.editage.cn] to improve the level of English.

Please also note the supplement to this comment:
https://www.ocean-sci-discuss.net/os-2018-88/os-2018-88-AC1-supplement.zip

---

## Author Comment (AC2) · 13 Nov 2018

We thank the Reviewer for the comments, which have helped improve our manuscript.

As human activities continue to expand into deep seas, more requirements are needed for deep-sea search and recovery. Currently, ROVs and HOVs are the main forces for deep sea recovery. Due to the high cost and scarcity of ROVs and HOVs, the authors in this paper proposed a new type of deep-sea recovery system based on TV-grab in oceanography. The quite detailed design and its recovery process are described. Although the new system has not been implemented and tested yet, based on this reviewer's experience, this concept is feasible and can easily be implemented. Therefore, the present concept is of some practical value and to those who have the TV-grabber

but lack of ROV and HOV, they can use TV-grabber to carry out the recovery task in an emergency situation. The paper is in overall quite comprehensive and well presented.

Response: We thank the Reviewer for the positive and detailed comments.

In terms of the originality, it is really not so much and in terms of the economic benefit, it is also very limited since TV-grabber is a type of ROV and their operational costs are more or the less the same.

Response: We thank the Reviewer for the comments, however, we do not agree completely with the Reviewer on this point. (1) A TV-grabber is not an ROV. There is a significant difference between them in deep sea exploration. Only deep-sea Heavy Work-Class ROVs with a tether management system (TMS) can be used for deep sea recovery. This type of ROV is very complex, and it requires a special technical maintenance team, and has a high diving cost [1-3]. On the one hand, a TV-grabber is much simpler, much easier to use, and it is very economical in deep-sea exploration [4,5].

(2) ROVs rely on their manipulators to grab targets, and the load of an ROV is limited, so it can only lift lightweight objects in the ocean [2,3]. On the other hand, most TV-grabs can sample up to 1000 kg or more at a time [4,5], and the maximum weight of our new deep-sea recovery system for lifting a lost target is up to 1000 kg in water.

(3) Our new deep-sea recovery system is based on the design idea and working mode of TV-grabbers, but it is not a TV-grabber. It is specially designed for deep sea recovery.

Compared with ROVs, our new deep-sea recovery system can provide low-cost and rapid deep-sea recovery. Thus, our new deep-sea recovery system can bring much more benefits.

[1] https://en.wikipedia.org/wiki/Remotely_operated_underwater_vehicle.

[2] Martin, A.Y.: Unmanned maritime vehicles: Technology evolution and implications, Marine Technology Society Journal, 47, 72-83, 2013.

[3] Schilling, T.: 2013 state of ROV technologies, Marine Technology Society Journal, 47, 69-71, 2013.

[4] http://www.ifm-geomar.de/

[5] Clark, M. R., Consalvey, M., Rowden, A. A.: Biological sampling in the deep sea, Wiley-Blackwell, New Jersey, 207-227pp., 2016.

In terms of the technical contents, this reviewer thinks it is not ocean science scope but ocean technology scope. Therefore, it is recommended for rejection in this journal but the authors are suggested to submit to the technology scope journal.

Response: We thank the Reviewer for the comments, however, we do not agree completely with the Reviewer on this point. On behalf of the NHESS Editorial Board, Natascha Töpfer encouraged us to resubmit our manuscript to Ocean Science on July 18, 2018. In her email, she stated, "You are encouraged to consider a resubmission of your manuscript to a related journal: https://editor.copernicus.org/OS/transfer/nhess-2018-188". The website section of Ocean Science detailing subject areas clearly shows that "The journal covers instrument development, in situ observations, remote sensing, data assimilation, laboratory, and numerical and theoretical studies", and "The coverage of the journal is worldwide and includes the deep ocean, the shelf seas, and inland seas, now, in the past, and the future." Thus, although our manuscript is not a pure ocean science research, such as ocean currents and eddies, it is still in line with the Journal's subject areas.

Some other minor corrections and suggestions are given in the attached PDF file.

Response: Following the comments in the PDF file, we have revised the manuscript as listed below. Page 1 Line 14: Corrected. Page 1 Line 15: please give some examples for the difficulties. "(such as the requirement of a special technical maintenance and protection team, very high diving cost)" has been added. Page 2 Line 3: Corrected. Page 2 Line 5: Corrected. Page 2 Line 9: Here, several more earlier examples are

given, especially the recovery of H-Bomb and the sunk ALVIN submersible. "Alvin spent three months searching for the unexploded H-bomb on the Mediterranean Sea floor before locating it and enabling it to be recovered by an ROV into in 1966." has been added. Page 2 Line 22: Corrected. Page 3 Line 8: Corrected. Page 4 Line 7: please also give some examples and references here, e.g. Deepsea challenge expedition. "For example, China has lost its lander system in COMRA 37th Cruise on May 3, 2016. The reason is that the float balls were broken by the ship propeller, causing the buoyancy to be less than gravity and sinking into the sea floor (Cruise Site Command, 2016)." has been added. "Cruise Site Command: The site report of the COMRA 37th Cruise, Xiangyanghong 09, 2016." has been added to References. Page 5 Line 9: Corrected. Page 7 Line 27: Corrected.

Please also note the supplement to this comment:
https://www.ocean-sci-discuss.net/os-2018-88/os-2018-88-AC2-supplement.zip

---

## Referee Comment (RC3) · Anonymous Referee #2 · 30 Nov 2018

The authors describe the idea of creating a new deep sea recovery system based on a TV-grab. The work is presented well enough, with a sufficient English, but the proposed topic is not in line with Ocean Science aims. Furthermore, no experimental results are presented. The conclusions affirm that the proposed system is better than the other technologies taken into consideration (i.e. HOV / ROV), without demonstrating it. I do not understand what the authors mean on page 5, line 4: "To map the seafloor topography at the depths of several or even tens of kilometers....". Do the authors speak about depth or they are speaking about the surface of the investigated seabed? If they talk about depth it is a very serious mistake.

---

## Referee Comment (RC4) · Anonymous Referee #3 · 12 Dec 2018

The paper describes traditional methods for the search and recovery of objects lost at the deep sea and it also describes a new system for this search-recovery problem. This new system seems interesting in practice; however, some points should be discussed in the study:

1. The sections 2, 3 and 4 basically describe the objects that are typically lost at the deep sea and the methods for searching and rescuing them. These sections are long and do not include any new scientific contribution. They could thus be reduced and merged into the introduction. Further, some parts of these sections seem to lack reference, for example: line 9-13 on page 2, line 5-10 on page 3, line 1 on page 4, line 21-23 on page 6, and line 14-19 on page 7.

2. The objective of the study could be clearer at the end of the Introduction (lines 22-25

on page 1). Especially, the last sentence (line 25 on page 1) lacks precision.

3. The new system proposed in the publication is very well described in section 5. However, even though it has not been tested in the field yet, some deeper discussion regarding the outcomes of its use should be presented in this study (costs, time of operation, possible damages on the lost object during grabbing or others). Further, a detailed scientific methodology should be considered to substantiate the discussion.

4. The recovery system seems to be designed to be directly connected to the winch of the vessel. In this case, the motion of the vessel would be coupled to the recovery system, inducing possible high dynamic effects on the system. How would these effects interfere with the searching process and with the controllability of the system (especially during the grabbing phase)?

5. Subsea lifting operations are ruled by some international standards, such as DNV-OS-H205, DNV-OS-H206, and DNV-RP-H103. How does the operation described in page 14 (especially lines 17-26) comply with these regulations?

---

## Author Comment (AC3) · 26 Dec 2018

We thank the Reviewer for the comments, which have helped us improve our manuscript. Our responses are in blue.

The authors describe the idea of creating a new deep sea recovery system based on a TV-grab. The work is presented well enough, with a sufficient English, but the proposed topic is not in line with Ocean Science aims. Response: We thank the Reviewer for this comment; however, we do not agree completely with the Reviewer on the point that "the proposed topic is not in line with Ocean Science aims". On behalf of the NHESS Editorial Board, Natascha Töpfer encouraged us to resubmit our manuscript to Ocean Science on July 18, 2018. In her email, she stated, "You

are encouraged to consider a resubmission of your manuscript to a related journal: https://editor.copernicus.org/OS/transfer/nhess-2018-188". The website section of Ocean Science detailing subject areas clearly shows that "The journal covers instrument development, in situ observations, remote sensing, data assimilation, laboratory, and numerical and theoretical studies", and "The coverage of the journal is worldwide and includes the deep ocean, the shelf seas, and inland seas, now, in the past, and the future." Thus, although our manuscript is not a pure ocean science research, such as ocean currents and eddies, we believe it is still in line with the Journal's subject areas. Furthermore, no experimental results are presented. Response: We thank the Reviewer for this comment. (1) In accordance with the Editorial opinions, we state that the new system described in this paper has yet to be tested in the Abstract and Section 5.3. (2) Our new deep-sea recovery system is based on the design idea and working model of TV- grab. Even though our new deep-sea recovery system is not tested, TV-grab is widely used in oceanography. (3) As shown in Figure 3, most of the onboard parts already exist on modern integrated ships and only a few units need to be added. Thus, the new deep-sea recovery system can work well. The next step is to test the new deep-sea recovery system in a practical application. The conclusions affirm that the proposed system is better than the other technologies taken into consideration (i.e. HOV / ROV), without demonstrating it. Response: We thank the Reviewer for this comment. (1) Only deep-sea Heavy Work-Class ROVs with a tether management system (TMS) can be used for deep sea recovery. This type of ROV is very complex, requires a special technical maintenance team and has a high diving cost [1-3]. On the other hand, a TV-grabber is much simpler, much easier to use, and it is very economical in deep-sea exploration [4,5]. (2) ROVs rely on their manipulators to grab targets, and the load-handling capacity of an ROV is limited, so it can only lift lightweight objects in the ocean [2,3]. On the other hand, most TV-grabs can sample up to 1000 kg or more at a time [4,5], and the maximum weight of our new deep-sea recovery system for lifting a lost target is up to 1000 kg in water. (3) Our new deep-sea recovery system is based on the design concept and working

model of TV- grabbers. It is specially designed for deep-sea recovery. Compared with ROVs, our new deep-sea recovery system can provide low-cost and rapid deep- sea recovery. Thus, our new deep-sea recovery system is better than ROVs/HOVs for this application. [1] https://en.wikipedia.org/wiki/Remotely_operated_underwater_vehicle. [2] Martin, A.Y.: Unmanned maritime vehicles: Technology evolution and implications, Marine Technology Society Journal, 47, 72-83, 2013. [3] Schilling, T.: 2013 state of ROV technologies, Marine Technology Society Journal, 47, 69-71, 2013. [4] http://www.ifm-geomar.de/ [5] Clark, M. R., Consalvey, M., Rowden, A. A.: Biological sampling in the deep sea, Wiley- Blackwell, New Jersey, 207-227pp., 2016. I do not understand what the authors mean on page 5, line 4: "To map the seafloor topography at the depths of several or even tens of kilometers....". Do the authors speak about depth or they are speaking about the surface of the investigated seabed? If they talk about depth it is a very serious mistake. Response: We thank the Reviewer for this comment. To clarify our meaning: The deep sea is the lowest layer in the ocean, existing below the thermocline, at a depth of 1800 m or more. Deep sea areas with a depth of more than 1800 m account for 84% of the ocean area. To map the deep sea (>1800 m) seafloor topography, a large-area walking survey can be performed with a shipborne deep-sea multi-beam sounding system to obtain relatively accurate data.

Please also note the supplement to this comment:
https://www.ocean-sci-discuss.net/os-2018-88/os-2018-88-AC3-supplement.pdf

---

## Author Comment (AC4) · 26 Dec 2018

We thank the Reviewer for the comments, which have helped us improve our manuscript. Our responses are in blue.

The paper describes traditional methods for the search and recovery of objects lost at the deep sea and it also describes a new system for this search-recovery problem. This new system seems interesting in practice; however, some points should be discussed in the study: Response: We thank the Reviewer for this positive and detailed comments. 1. The sections 2, 3, and 4 basically describe the objects that are typically lost at the deep sea and the methods for searching and rescuing them. These sections are long and do not include any new scientific contribution. They

could thus be reduced and merged into the introduction. Further, some parts of these sections seem to lack reference, for example line 9-13 on page 2, line 5-10 on page 3, line 1 on page 4, line 21-23 on page 6, and line 14-19 on page 7. Response: We thank the Reviewer for this comment, however, we do not agree completely with the Reviewer on this point. Section 2, 3 and 4 give the basis of the new deep-sea recovery system proposed in Section 5. All these sections constitute an organic whole of this paper. (1) Section 2 presents a general description of deep-sea search and recovery objects. From this section, one can obtain a general idea of deep-sea search and recovery objects, such as whether their location is known or not, their weight in water, their dimensions, and so on. These are needed for the designation of a new deep-sea recovery system in Section 5. (2) Section 3 gives the basic methods and equipment for deep-sea search. Before recovery and salvage, we should obtain the accurate position of the object. According to the accuracy of the object position, the new deep-sea recovery system proposed in Section 5 uses 3D real-time imaging sonar for early detection of salvage objects. (3) Section 4 analyzes the deep-sea recovery process and typical recovery cases based on operational underwater robots. It shows that there are many difficulties in practical application, such as the requirement for a special technical maintenance and protection team, and very high diving costs. Therefore, there is an urgent need for a low-cost deep-sea recovery system which proposed in Section 5. Furthermore, 8 references are added. [1] 1966 Palomares B-52 crash: https://en.wikipedia.org/w/index.php?title=1966_Palomares_B-52_crash&oldid=873145370, last access: 18 December 2018. [2] Air France Flight 447, https://en.wikipedia.org/w/index.php?title=Air_France_Flight_447&oldid=873450197, last access: 18 December 2018. [3] Autonomous Benthic Explorer, http://www.whoi.edu/main/ABE, last access: 18 December 2018. [4] Echoscope, https://www.codaoctopus.com/products/3d/echoscope, last access: 18 December 2018. [5] 370, https://en.wikipedia.org/w/index.php?title=Malaysia_Airlines_Flight_370&oldid=872787830, last access: 18 December 2018. [6] Nereus (underwater vehicle), https://en.wikipedia.org/w/index.php?title=Nereus_(underwater_vehicle)&oldid=853247050,

last access: 18 December 2018. [7] Soviet submarine K-278 Komsomolets, https://en.wikipedia.org/w/index.php?title=Soviet_submarine_K-278_Komsomolets&oldid=861297786, last access: 18 December 2018. [8] Space Shuttle Challenger disaster, https://en.wikipedia.org/w/index.php?title=Space_Shuttle_Challenger_disaster&oldid=873446304, last access: 18 December 2018. 2. The objective of the study could be clearer at the end of the Introduction (lines 22-25 on page 1). Especially, the last sentence (line 25 on page 1) lacks precision. Response: We thank the Reviewer for this comments. The part in question is rewritten as follows: The remainder of this paper is organized as follows. Section 2 presents a general description of deep-sea search and recovery objects. Section 3 gives the basic methods and equipment for deep-sea search. In Section 4, we analyze the deep-sea recovery process and typical recovery cases based on operational underwater robots. In Section 5, based on the design concept and working model of TV-grab in oceanographic studies, we propose a new type of deep-sea recovery system to perform the recovery process. Finally, we summarize this paper in Section 6. 3. The new system proposed in the publication is very well described in section 5. However, even though it has not been tested in the field yet, some deeper discussion regarding the outcomes of its use should be presented in this study (costs, time of operation, possible damages on the lost object during grabbing or others). Further, a detailed scientific methodology should be considered to substantiate the discussion. Response: We thank the Reviewer for this positive and detailed comments. "5.4 Discussion" is added as follows: 5.4 Discussion The new deep-sea recovery system is designed based on the concept of the TV-grab used in oceanographic studies. TV-grab does not need a technical support team in the practical application (Clark et al., 2016). The working model of the new deep-sea recovery system is similar to that of TV-grab. It only requires two or three people to operate. The cost of the new deep-sea recovery system in a practical application is much lower than the existing deep-sea recovery systems that are based on the deep-sea operation of ROVs and HOVs. Furthermore, as shown in Figure 3, most of

the onboard parts already exist on modern integrated ships and only a few units need to be added. Thus, the cost of building the new deep-sea recovery system is also very low. The new deep-sea recovery system is easy to launch and retrieve like TV-grab. It can operate in the ocean bottom for a long time. The docking/gripping system of the new deep-sea recovery system includes an actuator and drive mechanism that supports repeated opening and closing for handling salvage objects. If no dynamic positioning is available, the ship centers on the position of the salvage object and floats at low speed. The onboard operator can see the actuator and the salvage object from the camera and choose the appropriate part of the salvage object to grip. However, the buoyancy material or sensors may be damaged when the actuator grips the salvage object. 4. The recovery system seems to be designed to be directly connected to the winch of the vessel. In this case, the motion of the vessel would be coupled to the recovery system, inducing possible high dynamic effects on the system. How would these effects interfere with the searching process and with the controllability of the system (especially during the grabbing phase)? Response: We thank the Reviewer for this comments. (1) To minimize the motion effect of the ship, the ship should have a dynamic positioning system. If no dynamic positioning is available, the ship centers on the position of the salvage object and floats at low speed. (2) The recovery operation should be operated under level 4 sea conditions. Like TV-grab, operating in high-level sea condition will be very dangerous. In this tentative study, we only consider good sea conditions. High sea condition is beyond the scope of this study, but we will consider it in our future studies. 5. Subsea lifting operations are ruled by some international standards, such as DNV- OS-H205, DNV-OS-H206, and DNV-RP-H103. How does the operation described in page 14 (especially lines 17-26) comply with these regulations? Response: We thank the Reviewer for this comment. (1) Because the new deep-sea recovery system is similar to TV-grab, we use the working model of TV-grab for reference when describing the recovery process in Section 5.3. (2) In this tentative study, we refer to relevant standards, such as "General Rules for Diving and Underwater Operations", and "Requirements for Unmanned Remote Control

Submersible Cooperative Diving Operations". (3) In the future, we will improve this new deep-sea recovery system to meet international standards. We will give serious consideration to the international standards, such as DNV- OS-H205, DNV-OS-H206, and DNV-RP-H103.

Please also note the supplement to this comment:
https://www.ocean-sci-discuss.net/os-2018-88/os-2018-88-AC4-supplement.pdf
* * *

---

## Author Comment (AC7) · 31 Dec 2018

**Response letter**

**We thank the Reviewer for the comments, which have helped improve our manuscript. Our responses are in blue.**

I am happy with the reply.

**Response:** We thank the Reviewer for the positive comment.